

# Future climate-driven habitat loss and range shift of the Critically Endangered whitefin swellshark (*Cephaloscyllium albipinnum*)

Kerry Brown and Robert Puschendorf

School of Biological and Marine Sciences, University of Plymouth, Plymouth, Devon, United Kingdom

Corresponding author
Robert Puschendorf,
robert.puschendorf@plymouth.ac.uk

## ABSTRACT

Climate change is driving many species to shift their geographical ranges poleward to maintain their environmental niche. However, for endemic species with restricted ranges, like the Critically Endangered whitefin swellshark (*Cephaloscyllium albipinnum*), endemic to southeastern Australia, such dispersal may be limited. Nevertheless, there is a poor understanding of how *C. albipinnum* might spatially adjust its distribution in response to climate change or whether suitable refugia exist for this species in the future. Therefore, to address this gap, this study utilised maximum entropy (MaxEnt) modelling to determine the potential distribution of suitable habitat for *C. albipinnum* under present-day (2010–2020) climate conditions and for future conditions, under six shared socioeconomic pathways (SSP1-1.9, SSP1-2.6, SSP2-4.5, SSP3-7.0, SSP4-6.0 and SSP5-8.5) for the middle (2040–2050) and end (2090–2100) of the century. Under present-day conditions (2010–2020), our model predicted a core distribution of potentially suitable habitat for *C. albipinnum* within the Great Australian Bight (GAB), with benthic primary productivity and surface ocean temperature identified as key distribution drivers. However, under all SSP scenarios, future projections indicated an expected range shift of at least 72 km, up to 1,087 km in an east-southeast direction towards Tasmania (TAS). In all future climate scenarios (except SSP1-1.9 by 2100), suitable habitat is expected to decline, especially in the high-emission scenario (SSP5-8.5), which anticipates a loss of over 70% of suitable habitat. Consequently, all future climate scenarios (except SSP1-1.9 by 2100) projected a decrease in suitable habitat within a currently designated marine protected area (MPA). These losses ranged from 0.6% under SSP1-1.9 by 2050 to a substantial 89.7% loss in coverage under SSP5-8.5 by 2100, leaving just 2.5% of suitable habitat remaining within MPAs. With *C. albipinnum* already facing a high risk of extinction, these findings underscore its vulnerability to future climate change. Our results highlight the urgency of implementing adaptive conservation measures and management strategies that consider the impacts of climate change on this species.

## INTRODUCTION

A widely recognised consequence of climate change is the occurrence of geographical range shifts, whereby species are anticipated to undergo alterations in their spatial distribution, thereby preserving their environmental niche (*Chen et al., 2011*; *Walther et al., 2002*). One approach used to understand this phenomenon is the application of species distribution modelling (SDM). By integrating known species occurrence (or abundance) data with environmental variables, SDMs can predict the potential distribution of suitable habitat across space and time (*Elith & Leathwick, 2009*; *Miller, 2010*). Despite a bias in the SDM literature towards the terrestrial realm (*Robinson et al., 2011*), recent years have seen a growing body of evidence documenting range shifts in the marine environment in response to changing oceanographic conditions (*Melo-Merino, Reyes-Bonilla & Lira-Noriega, 2020*; *Robinson et al., 2017*). A diverse array of marine taxa, including plankton (*e.g.*, *Benedetti et al., 2021*), demersal fishes (*e.g.*, *Dulvy et al., 2008*), and marine mammals (*e.g.*, *Chambault et al., 2022*; *Fu et al., 2021*) are forecasted to shift poleward and/or into deeper water in response to climate change. Notably, despite ocean regions warming slower than land (*Intergovernmental Panel on Climate Change, 2023*), *Lenoir et al. (2020)* found that marine species are shifting their distributions poleward at an average rate six times that of terrestrial species. This can be attributed, in part, to fewer physical barriers within the marine environment (as opposed to terrestrial habitats), allowing for greater dispersal and colonisation abilities in the ocean if suitable habitat is available (*Poloczanska et al., 2013*). However, for species with restricted ranges, such as endemic species, opportunities to shift their range in response to climate change may be limited (*Kitchel et al., 2022*), with climate-related extinction risk more than twice as high for endemics than for native species (*Manes et al., 2021*).

Among the marine taxa particularly vulnerable to climate-driven shifts are sharks. As ectothermic species (except Lamnid sharks; *Carey et al., 1971*), sharks rely primarily on the surrounding environment to regulate their body temperature, which in turn directly influences vital metabolic and physiological functions such as digestion, growth, and reproduction (*Bernal et al., 2012*). Furthermore, sharks are already facing a high extinction risk due to overfishing (*Dulvy et al., 2021*), owing to their life history characteristics (*i.e.*, late maturity, low fecundity, long lifespan, low natural mortality) (*Camhi et al., 1998*), traits that can reduce their capacity to recover once populations are depleted (*Cortés, 1998*; *Finucci et al., 2024*). Australia is one of the most diverse regions for sharks globally, with around 180 recognised species, of which approximately 70 are unique to Australian waters (*Last & Stevens, 2009*). Of these endemic species, an estimated 5.8% are threatened with extinction, while 27.7% are classified as Data Deficient by the International Union for Conservation of Nature Red List (*International Union for Conservation of Nature, 2023*). Notably, the whitefin swellshark (*Cephaloscyllium albipinnum*), classified as Critically Endangered with its population facing an ongoing decline (*Pardo et al., 2019*), stands out as a species of critical concern.

*C. albipinnum* is a benthic catshark endemic to southeastern Australia, found at depths of 125 to 555 m on the outer continental shelf and upper continental slope (*Ebert, Dando & Fowler, 2021*). Despite belonging to the most speciose family (Scyliorhinidae; *Ebert, Dando & Fowler, 2021*), *C. albipinnum* remains poorly understood, with ecological information remaining scarce, particularly on habitat utilisation and movement patterns. Previous studies have, however, described members of the Scyliorhinidae family as non-migratory, slow-moving animals exhibiting an anguilliform mode of swimming (*Ferragut-Perello et al., 2024*; *Sternes & Shimada, 2020*; *West, Curtin & Woledge, 2022*). Tagging studies have indicated a high degree of site fidelity among catsharks. For instance, *Rodríguez-Cabello et al. (2004)* found that 70% of recaptured *Scyliorhinus canicular* did not travel more than 24 km, with a maximum distance record of 256 km. Similarly, *Awruch et al. (2012)* found that despite *Cephaloscyllium laticeps* exhibiting more extensive movements of up to 300 km, most recaptured individuals moved less than 10 km from their release site. Catsharks, in general, are sedentary and have been found to prefer hard substrates, such as small rocky crevices and caves, where individuals find refuge, resting motionless for extended periods, either alone or in aggregations (*Sims, Nash & Morritt, 2001*; *Sims et al., 2005*). In addition, nocturnal activity patterns have been observed in several *Cephaloscyllium* species, namely, *Cephaloscyllium ventriosum* (*Nelson & Johnson, 1970*), *C. laticeps* (*Awruch et al., 2012*) and *C. isabellum* (*Kelly et al., 2020*), indicating a preference for nighttime foraging. As opportunistic feeders, the diet of documented *Cephaloscyllium* species is diverse, often dominated by teleosts, crustaceans, and cephalopods (*Barnett et al., 2013*; *Horn, 2016*; *Taniuchi, 1988*).

Given its bottom-dwelling nature and limited movement capabilities, *C. albipinnum* is highly susceptible to frequent bycatch, notably from bottom longlines and trawlers (*Pardo et al., 2019*). Despite not being a targeted species, estimates suggest that over the past three generations (45 years), *C. albipinnum* has undergone a population reduction of more than 80% (*Pardo et al., 2019*). Although fishing pressure remains the primary threat to *C. albipinnum* (*Pardo et al., 2019*), climate change could exacerbate existing challenges for this species. The southeast and southwest of Australia are recognised as global warming 'hotspots', warming at rates almost four times the global average (*Hobday & Pecl, 2013*). Furthermore, warming trends are projected to continue, with ocean surface temperatures increasing by 0.86 to 2.89 °C by the end of the century, depending on greenhouse gas emission levels (*Intergovernmental Panel on Climate Change, 2023*). Australia has no specific conservation or management measures in place for *C. albipinnum* (*Pardo et al., 2019*). However, general conservation measures for other deepwater sharks off southeastern Australia, including spatial closures for gulper sharks (Centrophoridae; *Australian Fisheries Management Authority, 2022*) and existing marine protected areas (MPAs), could contribute to the conservation of *C. albipinnum* through indirect benefits, such as habitat protection and reduced fishing pressure (*Albano et al., 2021*; *Speed, Cappo & Meekan, 2018*). However, climate-driven range shifts beyond MPAs may increase its vulnerability. Incorporating habitat predictions into fisheries management could dynamically identify and protect shifting critical habitats, complementing static MPAs and ensuring adaptive conservation strategies.

While some empirical evidence of climate-driven shifts in the distribution of shark species exists, these studies are often concentrated on commercially valuable groups (*e.g.*, *Birkmanis et al., 2020*; *Diaz-Carballido et al., 2022*) or those which are highly mobile (*e.g.*, *Hammerschlag et al., 2022*; *Womersley et al., 2024*). This leaves a critical knowledge gap regarding how endemic and less-studied species such as *C. albipinnum* might spatially adjust their distribution in response to climate change and whether suitable refugia exist for such species in the future. Therefore, this study aims to assess the potential effects of climate change on the Critically Endangered whitefin swellshark (*C. albipinnum*). Specifically, we employ SDM to assess suitable habitat under six shared socioeconomic pathways (SSPs) (SSP1-1.9, SSP1-2.6, SSP2-4.5, SSP3-7.0, SSP4-6.0 and SSP5-8.5) for the middle (2050) and end (2100) of the century. Based on model projections, we aim to (1) estimate the current distribution of *C. albipinnum*, (2) estimate the future distribution of *C. albipinnum* under various climate change scenarios, and (3) evaluate the extent to which currently designated MPAs provide coverage for both the current and future distribution of *C. albipinnum*.

## MATERIALS AND METHODS

This study assessed the current and future suitable habitat for *C. albipinnum* using the maximum entropy method (MaxEnt). MaxEnt applies the maximum entropy principle (*i.e.*, most spread out or closest to uniform) to relate presence-only data to environmental factors to estimate a species' potential geographical distribution and environmental tolerances (*Phillips, Anderson & Schapire, 2006*). It is the preferred tool for SDM, particularly in the marine environment (*Melo-Merino, Reyes-Bonilla & Lira-Noriega, 2020*), due to its efficiency, ease of use, and consistently strong performance (*Elith et al., 2006*; *Valavi et al., 2021*), even with sparse, irregularly sampled occurrence data, which are constraints that are often encountered for rare, elusive or threatened species (*e.g.*, *Noviello et al., 2021*), as well as from poorly accessible areas (*e.g.*, *Hernandez et al., 2008*). In addition, the continuous output allows for fine distinctions between the modelled suitability of different areas, with the flexibility to apply thresholds for binary predictions when necessary (*Phillips, Anderson & Schapire, 2006*).

### Occurrence data

All 486 available occurrence records for *C. albipinnum* were obtained from the *Global Biodiversity Information Facility (2024)*, *Ocean Biodiversity Information System (2024)*, and the *Atlas of Living Australia (2024)* repositories. However, due to the likely error and uncertainty introduced using species data amalgamated from several sources, records were manually cleaned and refined using the following steps. First, although occurrence records extended over 115 years, localities pre-dating 1965, along with records that did not include a collection date, were disregarded due to concerns regarding accuracy. Next, a visual inspection was conducted using Quantum Geographic Information System software (QGIS; v3.38.1; *QGIS Geographic Information System, 2024*), where occurrences located outside the recognised range of the species, as delineated by the Australian National Fish Expert Distribution (ANFED; *CSIRO Marine and Atmospheric Research, 2012*) and the

International Union for Conservation of Nature Red List (*International Union for Conservation of Nature, 2012*) (see Fig. S1), or those exceeding coastline boundaries (*e.g.*, occurring on land) as defined by Digital Earth Australia (v2.1.0; *Bishop-Taylor et al., 2021*), were excluded due to potential georeferencing errors, along with incomplete records lacking coordinates. After cleaning, the remaining 405 occurrence records comprised of 'preserved specimen' (39.5%), 'human observation' (58.8%) and 'material sample' (1.7%). Lastly, to reduce the effects of sampling bias and prevent model overfitting (*Boria et al., 2014*; *Kramer-Schadt et al., 2013*), duplicates were removed, and a spatial filter was applied in R (v4.4.1; *R Core Team, 2024*) using the '*spThin*' package (*Aiello-Lammens et al., 2015*). We assigned a 5.5 km radius (*i.e.*, one record per $0.05° × 0.05°$ pixel), consistent with the environmental predictors' resolution. Finally, 145 *C. albipinnum* occurrence records were retained for use in the final model (see Fig. S1).

## Environmental data

Initially, 14 environmental variables were considered as predictors to model suitable habitat for *C. albipinnum*. These variables were chosen based on their direct or indirect (*e.g.*, serving as a proxy for prey) relevance and availability (see Table S1). Oceanographic predictors included ocean temperature (°C), salinity (PSS), dissolved molecular oxygen (nmol m$^{-3}$), seawater velocity (m s$^{-1}$), primary productivity (nmol m$^{-3}$) and chlorophyll-a concentration (nmol m$^{-3}$). Given that *C. albipinnum* primarily inhabits benthic regions but can still be influenced by surface conditions (*Ebert, Dando & Fowler, 2021*), both the benthic (except chlorophyll-a concentration) and surface layers for these variables were obtained. Topographic predictors included bathymetry (m), slope (°) and substrate type. Substrate type was categorised into eight distinct classes: (1) biosiliceous marl and calcareous clay, (2) calcareous gravel, sand and silt, (3) calcareous ooze, (4) mud and calcareous clay, (5) mud and sand, (6) pelagic clay, (7) sand, silt and gravel with less than 50% mud, and (8) volcanic sand and grit. All environmental layers except substrate type were obtained from the Bio-ORACLE (v3.0; *Assis et al., 2024*) database. These layers were acquired at a spatial resolution of 0.05° (approximately 5.5 km at the equator) and represent the climatological average for the present-day (2010–2020) climate. Substrate data was obtained from the Commonwealth Scientific and Industrial Research Organisation (CSIRO) marine benthic substrate database (*Commonwealth Scientific and Industrial Research Organisation, 2015*). Substrate data was reclassified into a raster format using QGIS (v3.38.21; *QGIS Geographic Information System, 2024*) at a spatial resolution of approximately 0.05° (~5.5 km² per pixel) to match the other environmental variables.

Potential for model overfitting was reduced by analysing multicollinearity among the initial 14 candidate predictors using the following approach. First, we utilised the '*vifcor*' function from the '*usdm*' package (*Naimi et al., 2014*) in R (v4.4.1; *R Core Team, 2024*). The '*vifcor*' function first finds a pair of variables which has the maximum linear correlation ($|r| \geq 0.7$; *Dormann et al., 2012*) and excludes the variable with the greater Variance Inflation Factor (VIF). This procedure is repeated until no pair of variables with a high correlation coefficient remains. Next, we constructed an initial model using the MaxEnt software (v3.4.4; *Phillips, Anderson & Schapire, 2006*) with default parameters to

obtain a preliminary percentage contribution for each variable. Based on the average results of ten runs, environmental factors with a small contribution rate (≤1%) were excluded. Finally, we retained five environmental factors for modelling: slope, benthic primary productivity, surface ocean temperature, surface seawater velocity and surface salinity (see Table S1).

## Future projections

To predict potential future changes in the distribution of *C. albipinnum*, we considered two time periods, the middle (2040–2050) and the end (2090–2100) of the century, across six SSPs (SSP1-1.9, SSP1-2.6, SSP2-4.5, SSP3-7.0, SSP4-6.0 and SSP5-8.5). Ranging from the 'sustainability' scenario (SSP1-1.9), which aligns with the reduced greenhouse gas emissions targets of the Paris Agreement, to the 'fossil-fuelled development' scenario (SSP5-8.5), characterised by high emissions and low challenges to adaptation (*Riahi et al., 2017*). All future variables except slope were sourced from Bio-ORACLE (v3.0; *Assis et al., 2024*). Future projections were generated by averaging outputs from an ensemble of several Earth System Models (ESMs; ACCESS-ESM1-5, CanESM5, CESM2-WACCM, CNRM-ESM2-1, GFDL-ESM4, GISS-E2-1-G, IPSL-CM6A-LR, MIROC-ES2L, MPI-ESM1-2-LR, MRI-ESM2-0, UKESM1-0-LL) provided by the Coupled Model Intercomparison Project Phase 6 (CMIP6) (*Assis et al., 2024*). As a static topographic feature, slope remained consistent in future projections due to the lack of future estimates available in the Bio-ORACLE database.

## Calibration area

Several studies have indicated that the size of the calibration area and the environmental space it contains (*i.e.*, the background data used for calibration) have significant effects on SDM results (*Amaro et al., 2023*; *Luna, Peña-Peniche & Mendoza-Alfaro, 2024*). However, despite its significance, there is no consensus on how to select an appropriate calibration area, and several approaches have been utilised, such as buffers, polygons, or distances based on species dispersal abilities (*Rojas-Soto et al., 2024*). The criterion used to define the calibration area for *C. albipinnum* was based on ecological delimitation and dispersal abilities, using *C. laticeps* as a proxy (*Awruch et al., 2012*). Following methodologies as described by *Diaz-Carballido et al. (2022)*, we defined the calibration area for *C. albipinnum* using biogeographic units as delineated by Marine Ecoregions of the World (MEOW; *Spalding et al., 2007*). MEOW regions were selected if they contained at least one occurrence point and/or within the known range for *C. albipinnum* as outlined by the ANFED (*CSIRO Marine and Atmospheric Research, 2012*) and the IUCN Red List (*International Union for Conservation of Nature, 2012*). The selected MEOW regions (*n* = 7) were grouped to form the M area (see Fig. S1), representing the geographic regions accessible to *C. albipinnum* over time, consistent with the M region concept described by *Soberon & Peterson (2005)*. The '*sf*' (v1.0; *Pebesma, 2018*) and '*raster*' (v3.6; *Hijmans, 2023*) packages in R (v4.4.1; *R Core Team, 2024*) were utilised to mask the environmental layers to the defined M area.

## MODELLING

### Model calibration

Model calibration, the final model, future projections, as well as assessment of extrapolation risks were all conducted in R (v4.4.1; *R Core Team, 2024*) utilising the '*kuenm*' package (v1.1.10; *Cobos et al., 2019*), and the MaxEnt Java program (v3.4.4; *Phillips et al., 2017*). Prior to calibration, occurrence records were spilt randomly into 70–30% subsets for model calibration and internal testing, respectively. The default configuration provided by MaxEnt is not necessarily the most appropriate, and species-specific tuning of model parameters, such as feature class (FC) and regularisation multiplier (RM), has been found to improve the predictive accuracy and performance of MaxEnt models (*Anderson & Gonzalez, 2011*; *Shcheglovitova & Anderson, 2013*). Our approach tested eight RM values 0.5–4.0 (in increments of 0.5) and all 15 combinations of four FCs (linear (l), quadratic (q), product (p), and hinge (h)). The threshold FC was omitted to create biologically meaningful model interpretations and improve model performance (*Merow, Smith & Silander, 2013*). Candidate models were evaluated based on three criteria: (1) statistical significance ($P \leq 0.05$), based on partial receiver operating characteristic (partial ROC; *Peterson, Papeş & Soberón, 2008*), generated with 500 iterations and 50% of data for bootstrapping, (2) predictive performance, based on omission rates ($E = 5\%$), and (3) minimum complexity, evaluated using the Akaike Information Criterion (*Akaike, 1973*) corrected for small sample sizes (AICc; *Hurvich & Tsai, 1989*), specifically those with delta AICc values lower than two.

### Model construction and validation

We created the final model using the '*kuenm_mod*' function from the '*kuenm*' package (v1.1.10; *Cobos et al., 2019*), using the selected parameterisations, complete set of occurrences ($n = 145$) and selected environmental variables ($n = 5$). We produced ten replicates by bootstrap, using the log-log (cloglog) output, which gives a probability of occurrence estimate between 0 (low probability) and 1 (high probability). The other parameters were left at their default values: a maximum of 10,000 randomly generated background points (from within the M area) and 500 maximum iterations with a $10^{-5}$ convergence threshold. The final model was projected to create a present-day (2010–2020) and 12 future predictions under six SSPs (SSP1-1.9, SSP1-2.6, SSP2-4.5, SSP3-7.0, SSP4-6.0 and SSP5-8.5) for two time periods, the middle (2040–2050) and the end (2090–2100) of the century. Extrapolation and clamping were selected for future projections.

Model performance was evaluated using the average area under the ROC curve (AUC). AUC values are commonly interpreted using a general classifying system: fail (<0.6), poor (0.6–0.7), fair (0.7–0.8), good (0.8–0.9), and values above 0.9 representing excellent model performance (*Phillips et al., 2017*). To identify the percentage contribution for each environmental variable, we used the Jackknife function of MaxEnt (*Phillips et al., 2017*). Lastly, to assess the transferability of our model and potential extrapolation risks, we utilised the mobility-oriented parity (MOP) metric. Areas with higher extrapolative values indicate higher uncertainty; thus, caution is required when interpreting the likelihood of

species presence in such areas (*Owens et al., 2013*). Although the MOP analysis depicted regions of strict extrapolation (see Fig. S2), these regions occur outside the areas predicted as suitable habitat for *C. albipinnum* and are not considered a concern for this study.

### Binary predictions

The maximum training sensitivity plus specificity (MTSS = 0.3576) was used to delineate suitable and unsuitable habitats; values below the threshold were deemed unsuitable, and values above the threshold represented suitable habitat for *C. albipinnum*. This threshold was chosen as it is recommended as a conservative approach that minimises both commission and omission errors (*Liu, White & Newell, 2013*). Then, following a similar framework to one previously described by *Diaz-Carballido et al. (2022)*, changes to suitable habitat were classified into three categories: (1) areas of contraction (currently suitable but not in the future), (2) areas of expansion (currently unsuitable but suitable in the future), and (3) stable areas (suitable both currently and in the future). Percentage values were calculated for each category as the proportion relative to the suitable habitat area predicted by the model (area of category/current area * 100). The total change was calculated as the proportion of change between current and future predicted areas (future area–current area/current area * 100); negative values indicate a net loss in suitable habitat, and positive values represent a net gain.

Range shifts for *C. albipinnum* were analysed by reducing the suitable habitat for the current distribution and all future distributions into single centroids. We then used R packages '*sf*' (v1.0; *Pebesma, 2018*) and '*geosphere*' (v1.5; *Hijmans, 2022*) to calculate the distance (km) and direction (bearing degrees). Lastly, the overlap between currently designated MPAs and projected suitable habitat for *C. albipinnum* was calculated using currently designated MPA data for Australian State and Commonwealth waters obtained from the *Collaborative Australian Protected Areas Database (2020)*. The MPA shapefile was cropped only to include MPAs within the M area and then intersected with each binary prediction to calculate the area of suitable habitat inside and outside the MPAs. All spatial analysis described above was conducted in QGIS (v3.38.1; *QGIS Geographic Information System, 2024*).

## RESULTS

### Model accuracy and variable importance

Considering the 15 combinations of the four FCs and eight RMs, 120 candidate models were created for *C. albipinnum*. All candidate models presented statistical significance ($P < 0.05$), and of the 120 candidate models, 46 models met the omission rate criterion (≤5%). However, only one model had a delta AICc value ≤2. Therefore, only one model (M_4_F_h_Set_1) met the three evaluation criteria. The parameters selected as optimal were an RM value of four, using only hinge features (see Table S2). The performance of the final model for *C. albipinnum* was considered excellent, with a mean AUC score of 0.94 (±0.006).

Three environmental factors contributed 94.6% to the model prediction of *C. albipinnum*, with benthic primary productivity being the highest contributor (70.8%),

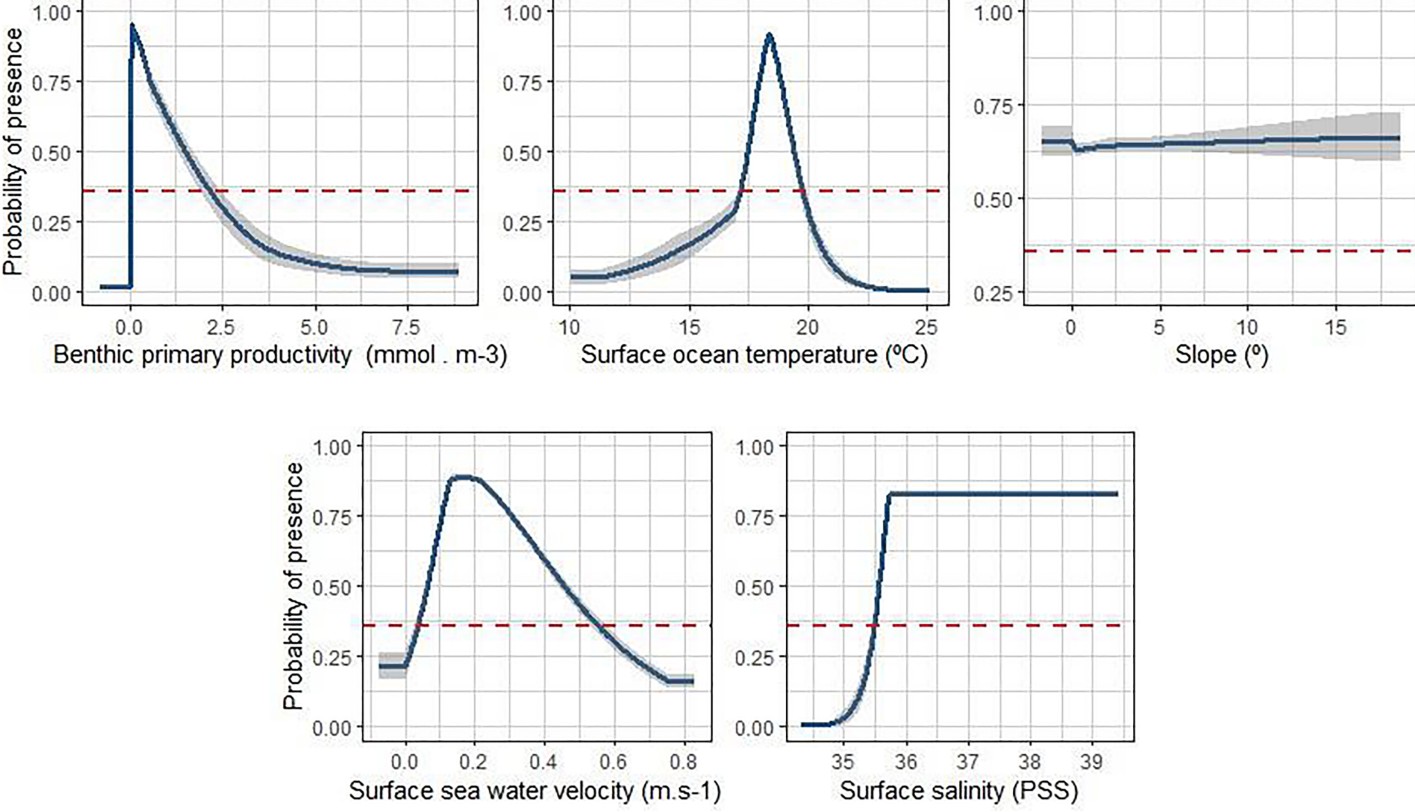

**Figure 1 Response curves between environmental variables and the probability of presence of the whitefin swellshark (*Cephaloscyllium albipinnum*).** Zero equals a low probability of presence and one equals a high probability. The blue curve indicates the mean response, and the grey margins are ± 1 standard deviation calculated over ten replicates. Values exceeding the binary threshold (0.3576; red dashed line) indicate suitable habitat conditions for *C. albipinnum*.

followed by surface ocean temperature (12.4%) and slope (11.7%). While the cumulative contribution of the two remaining variables accounted for 5% of the total contribution (see Fig. S3). The response curves demonstrate how variations in environmental factors affect the predicted probability of *C. albipinnum* presence (Fig. 1). Values exceeding the MTSS threshold (0.3576) indicate suitable habitat conditions for *C. albipinnum*. Environmental variable values were considered optimal when the response curves reached their maximum. Consequently, suitable habitat encompasses benthic primary productivity ranging from 0 to 2.2 mmol.m$^{-3}$, surface ocean temperatures between 17.2 and 19.8 °C, with an optimal temperature around 18.3 °C. Suitable surface seawater velocities between 0 and 0.6 m$^{-1}$, with an optimal velocity around 0.2 m$^{-1}$, surface salinities between 35.5 and 39.4 PSS, with an optimal salinity from 35.7 PSS and slopes between 0 and 18.7°.

## Current distribution

Figure 2 shows the predicted probability of presence and suitable habitat for *C. albipinnum* under present-day (2010–2020) climate conditions. The reclassified binary threshold (MTSS = 0.3576) estimated that under present-day (2010–2020) climate conditions, suitable habitat for *C. albipinnum* totalled 322,114 km², encompassing 14.4% of the total

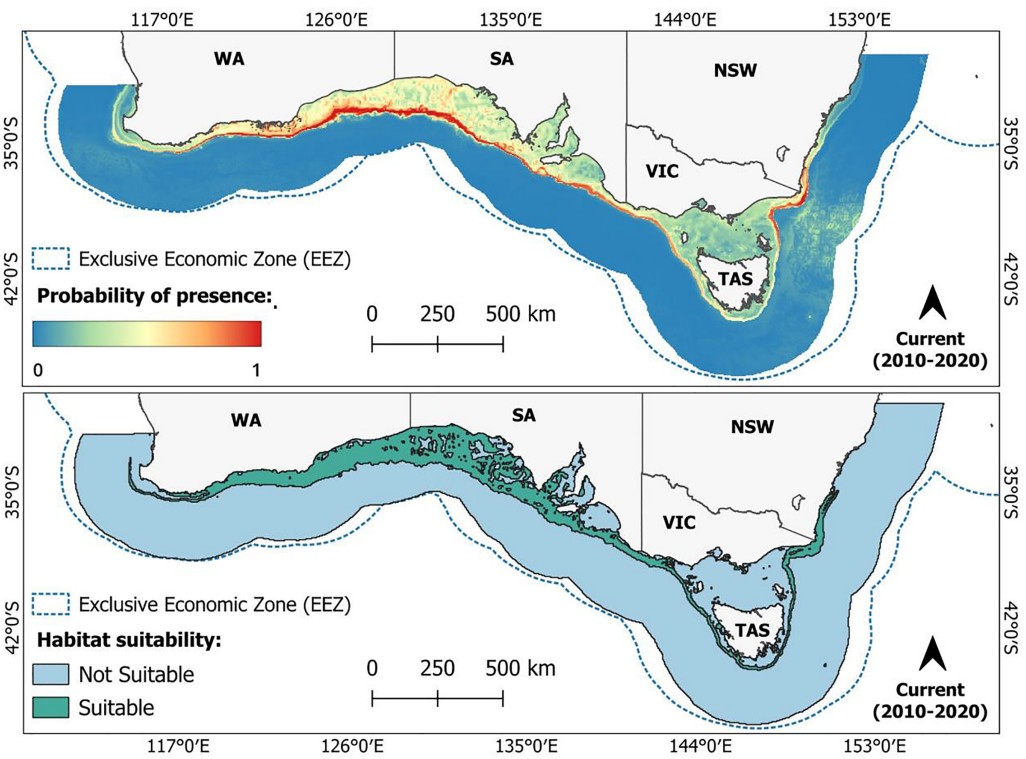

**Figure 2 Predicted current (2010–2020) probability of presence (top) and suitable habitat (bottom) for the whitefin swellshark (*Cephaloscyllium albipinnum*).** High probabilities of presence are indicated by warm colours, while low probabilities are represented by cool colours. Unsuitable areas for *C. albipinnum* are shown in blue, with suitable habitat depicted in green.

area (M area). The predicted suitable habitat for *C. albipinnum* spans across southern Australia from just above Cape Naturaliste, Western Australia (WA; ~33.2°S, 114.5°E) as far as Kiama Heights, New South Wales (NSW; ~34.7°S, 150.9°E), including Tasmania (TAS), but excluding the Bass Strait. The highest probability of presence was found primarily concentrated along the outer continental shelf and upper slope (Fig. 2). When compared to the known range of *C. albipinnum*, as delineated by the IUCN Red List (*International Union for Conservation of Nature, 2012*) and ANFED (*CSIRO Marine and Atmospheric Research, 2012*) (see Fig. S1), we observed slight overpredictions of approximately 850 km along the WA coastline, specifically from around Cape Arid, WA (~34.0°S, 123.6°E) to just above Cape Naturaliste, WA (~33.2°S, 114.5°E) along with small areas located within the Spencer Gulf and St. Vincent Gulf, South Australia (SA).

## Future distribution

For future projections, only one scenario predicted a net gain in suitable habitat for *C. albipinnum*. Under the SSP1-1.9 scenario, suitable habitat is expected to expand by the end of the century (2100), covering an area of 327,945 km², 1.8% larger than the current predicted distribution (Table 1). All remaining scenarios project a net loss, with a decline in suitable habitat expected to worsen as emissions increase and become more severe by

**Table 1 Area (km² ) and proportion (%) of habitat maintained, gained, and lost for the whitefin swellshark (*Cephaloscyllium albipinnum*).** Suitable area maintained, gained and lost under SSP1-1.9, SSP1-2.6, SSP2-4.5, SSP3-7.0, SSP4-6.0 and SSP5-8.5 by the middle (2040–2050) and the end of the century (2090–2100) relative to the current (2010–2020) distribution.

| Scenario | Suitable Habitat km² (%) | | | | |
|---|---|---|---|---|---|
| | Total Area | Maintained | Gained | Lost | Total Change (gain–loss) |
| Current | 322,114 | – | – | – | – |
| **2050** | | | | | |
| SSP1 1.9 | 319,415 | 294,540 (91.4) | 24,880 (7.7) | 27,565 (8.6) | −2,685 (−0.8) |
| SSP1 2.6 | 300,242 | 262,589 (81.5) | 37,664 (11.7) | 59,528 (18.5) | −21,864 (−6.8) |
| SSP2 4.5 | 280,954 | 234,545 (72.8) | 46,419 (14.4) | 87,574 (27.2) | −41,155 (−12.8) |
| SSP3 7.0 | 281,461 | 228,057 (70.8) | 53,413 (16.6) | 94,068 (29.2) | −40,655 (−12.6) |
| SSP4 6.0 | 289,283 | 243,277 (75.5) | 46,017 (14.3) | 78,844 (24.5) | −32,827 (−10.2) |
| SSP5 8.5 | 257,018 | 201,000 (62.4) | 56,027 (17.4) | 121,118 (37.6) | −65,091 (−20.2) |
| **2100** | | | | | |
| SSP1 1.9 | 327,945 | 301,943 (93.7) | 26,003 (8.1) | 20,174 (6.3) | 5,829 (1.8) |
| SSP1 2.6 | 272,900 | 217,734 (67.6) | 55,177 (17.1) | 104,388 (32.4) | −49,211 (−15.3) |
| SSP2 4.5 | 246,481 | 128,565 (39.9) | 117,926 (36.6) | 193,550 (60.1) | −75,624 (−23.5) |
| SSP3 7.0 | 166,682 | 73,406 (22.8) | 93,276 (29.0) | 248,714 (77.2) | −155,438 (−48.3) |
| SSP4 6.0 | 242,219 | 100,040 (31.1) | 142,187 (44.1) | 222,077 (68.9) | −79,890 (−24.8) |
| SSP5 8.5 | 90,178 | 45,990 (14.3) | 44,188 (13.7) | 276,123 (85.7) | −231,935 (−72.0) |

**Note:**
SSP, shared socioeconomic pathway.

the end of the century than in the 2050s (Table 1). By 2050, habitat contraction ranged from a slight 0.8% decrease under SSP1-1.9 to a 20.2% reduction under the high-emission SSP5-8.5 scenario (Table 1). By the end of the century (2100), the decrease in suitable habitat becomes more pronounced, ranging from a 15.3% loss under the SSP1-2.6 scenario to a substantial decline of 85.7% under the high emission scenario (SSP5-8.5), leaving just 90,178 km² (28%) potential suitable habitat remaining (Table 1).

Suitable habitat contraction under the lower emission scenarios begins in the coastal waters of WA, SA and NSW (Fig. 3; Figs. S4 and S5). As emissions increase, contraction shifts gradually towards the continental shelf. By the end of the century (2100), under the highest emission scenario, the entire predicted suitable habitat that falls within WA waters, a significant portion within SA waters, and an area spanning from Kiama Heights, NSW (~34.7°S, 150.8°E) to Hobart, TAS (~42.9°S, 148.6°E), are projected to become unsuitable (Fig. 3). Conversely, gains in suitable habitat were primarily observed along the Tasmanian continental shelf and in the Bass Strait (Fig. 3; Fig. S5). Smaller areas of habitat expansion were also identified within the Spencer Gulf, St. Vincent Gulf and Long Bay, SA (Fig. 3; see Figs. S4 and S5). Only 14.3% (45,990 km²) of the predicted present-day suitable habitat was maintained across all scenarios (Table 1). This region predominately spans the outer continental shelf, stretching from approximately Coorabie, SA (~34.0°S, 132.3°E) to Hobart, TAS (~42.9°S, 148.6°E) (Fig. 3).

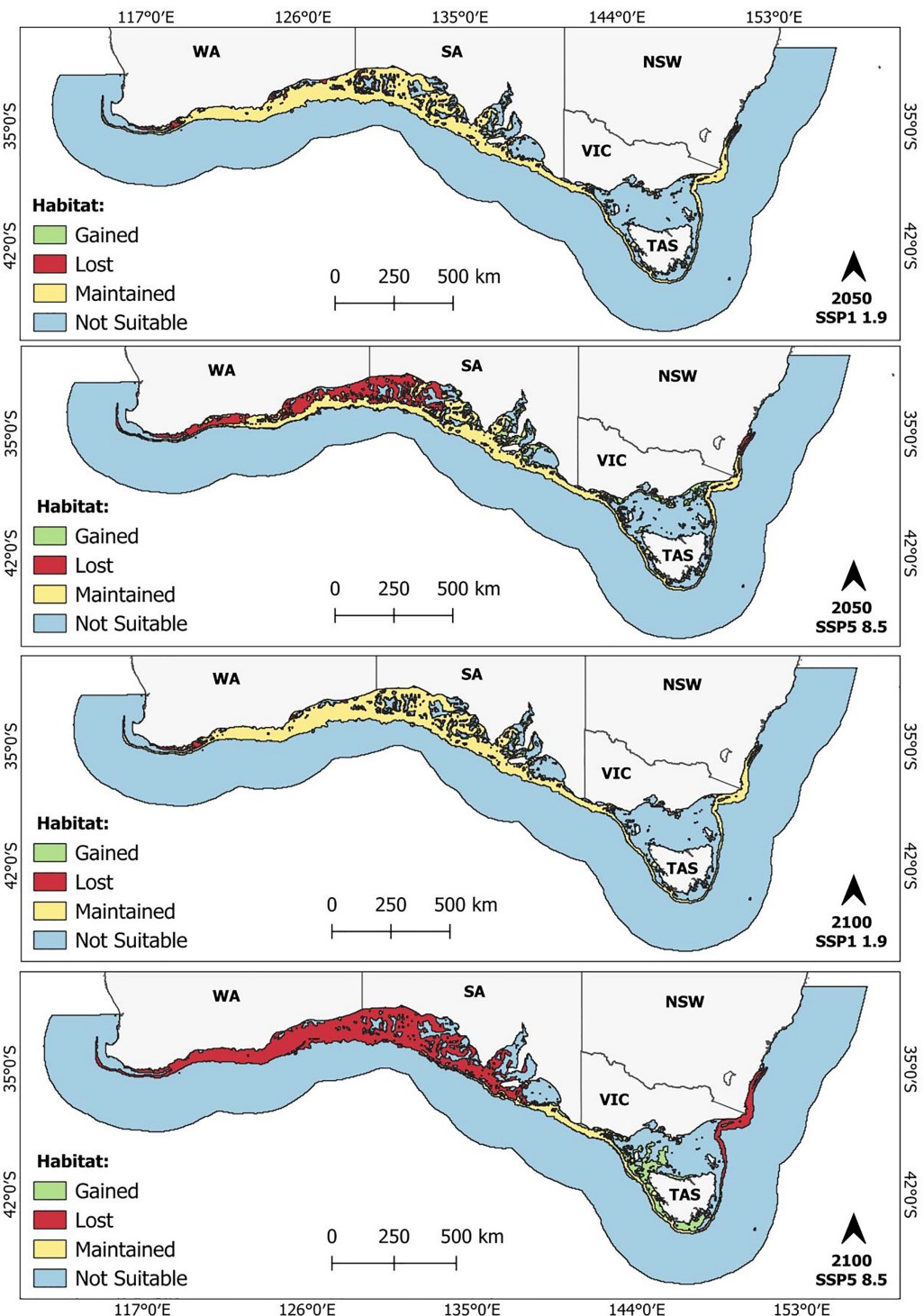

**Figure 3 Future predicted suitable habitat maintained (yellow), gained (green), and lost (red) for the whitefin swellshark (*Cephaloscyllium albipinnum*).** Included are scenarios SSP1-1.9 and SSP5-8.5 by the middle (2040–2050) (top two panels) and end of the century (2090–2100) (bottom two panels). Unsuitable areas for *C. albipinnum* are shown in blue. Binary threshold = 0.3576.

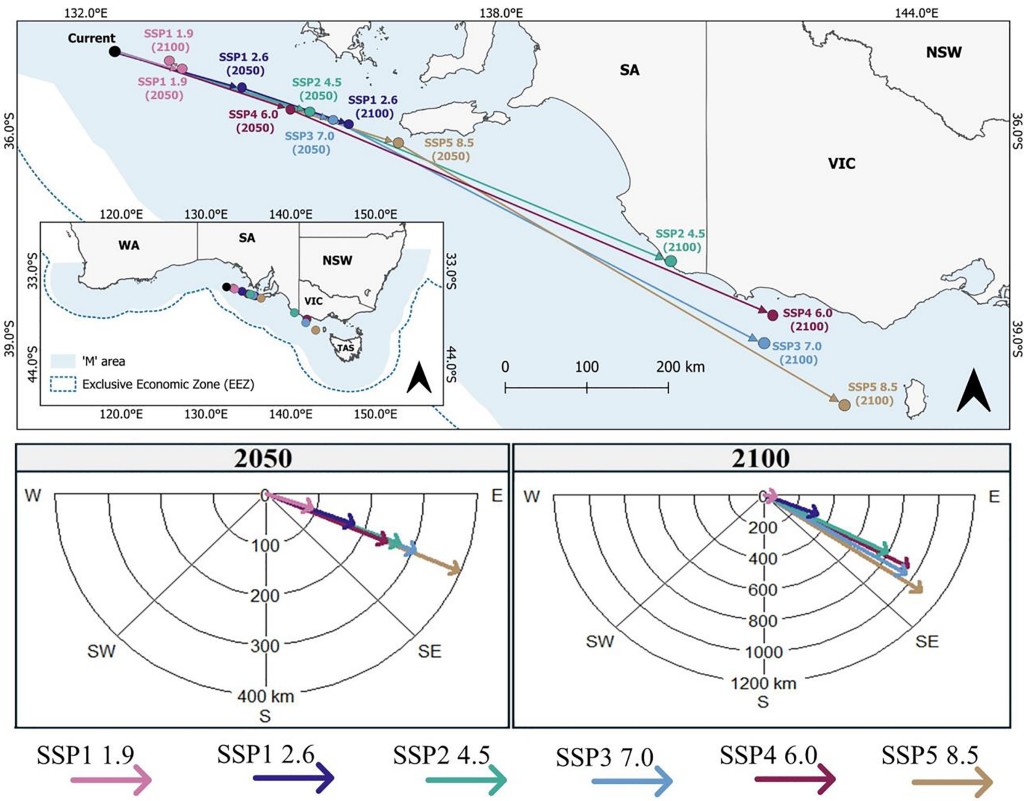

**Figure 4 Projected range shifts for the whitefin swellshark (*Cephaloscyllium albipinnum*) under SSP1-1.9, SSP1-2.6, SSP2-4.5, SSP3-7.0, SSP4-6.0 and SSP5-8.5 by the middle (2040–2050) and the end of the century (2090–2100).** The top panel shows the predicted geographic distribution of centroids under each SSP scenario, while the bottom panels illustrate changes in the direction and distance (km) of centroids between current and future projections for the middle (left) and end of the century (right). Binary threshold = 0.3576.     

All future predictions indicated that the core distribution (centroid) of suitable habitat for *C. albipinnum* would shift in an east-southeast direction from within the Great Australian Bight (GAB; 34.9°S, 132.4°E) towards TAS (Fig. 4). By the middle of the century (2050), this shift is projected to range from a minimum distance of 91.5 km under scenario SSP1-1.9 to a maximum distance of 396.4 km under SSP5-8.5 (Fig. 4). By the end of the century (2100), shifts ranged from 71.9 km to 1,086.7 km under scenarios SSP1-1.9 and SSP5-8.5, respectively (Fig. 4). Furthermore, by 2100, the core distribution of suitable habitat for *C. albipinnum* is predicted to shift beyond the SA border into waters off Victoria (VIC) under three scenarios, namely, SSP3-7.0 (39.1°S, 141.7°E), SSP4-6.0 (38.6°S, 141.9°E) and SPP5-8.5 (39.9°S, 142.9°E) (Fig. 4).

## Overlap with MPAs

Based on current MPA designations, 23.7% of the predicted present-day suitable habitat for *C. albipinnum* falls within an MPA, covering a total area of 76,379 km² (Table 2). Most of this habitat is located within MPAs that form part of the South-west Marine Parks Network, particularly the Great Australian Bight Marine Park (~21,665 km²) and the Western Eyre Marine Park (~17,933 km²) (Fig. 5). For future predictions, one scenario

**Table 2 Area (km²) and proportion (%) of potential suitable habitat for the whitefin swellshark (*Cephaloscyllium albipinnum*) that falls inside and outside marine protected areas (MPAs).** Suitable habitat predicted under SSP1-1.9, SSP1-2.6, SSP2-4.5, SSP3-7.0, SSP4-6.0 and SSP5-8.5 by the middle (2040–2050) and the end of the century (2090–2100) relative to the current (2010–2020) distribution.

| Scenario | Total Suitable Habitat km² | Suitable Habitat km² (%) | | |
|---|---|---|---|---|
| | | Outside MPAs | Inside MPAs | Change (Future Inside MPAs–Current Inside MPAs) |
| Current | 322,114 | 245,736 (76.3) | 76,379 (23.7) | – |
| **2050** | | | | |
| SSP1 1.9 | 319,415 | 243,525 (75.6) | 75,903 (23.6) | −476 (−0.6) |
| SSP1 2.6 | 300,242 | 230,739 (71.6) | 69,509 (21.6) | −6,870 (−9.0) |
| SSP2 4.5 | 280,954 | 220,206 (68.4) | 60,758 (18.9) | −15,621 (−20.5) |
| SSP3 7.0 | 281,461 | 221,937 (68.9) | 59,525 (18.5) | −16,854 (−22.1) |
| SSP4 6.0 | 289,283 | 225,796 (70.1) | 63,492 (19.7) | −12,887 (−16.9) |
| SSP5 8.5 | 257,018 | 207,032 (64.3) | 49,991 (15.5) | −26,388 (−34.5) |
| **2100** | | | | |
| SSP1 1.9 | 327,945 | 247,956 (77.0) | 79,992 (24.8) | 3,613 (4.7) |
| SSP1 2.6 | 272,900 | 215,828 (67.0) | 57,078 (17.7) | −19,301 (−25.3) |
| SSP2 4.5 | 246,481 | 212,578 (66.0) | 33,921 (10.5) | −42,458 (−55.6) |
| SSP3 7.0 | 166,682 | 152,297 (47.3) | 14,379 (4.5) | −62,000 (−81.2) |
| SSP4 6.0 | 242,219 | 215,741 (67.0) | 26,490 (8.2) | −49,889 (−65.3) |
| SSP5 8.5 | 90,178 | 82,275 (25.5) | 7,904 (2.5) | −68,475 (−89.7) |

Note:
SSP, shared socioeconomic pathway.

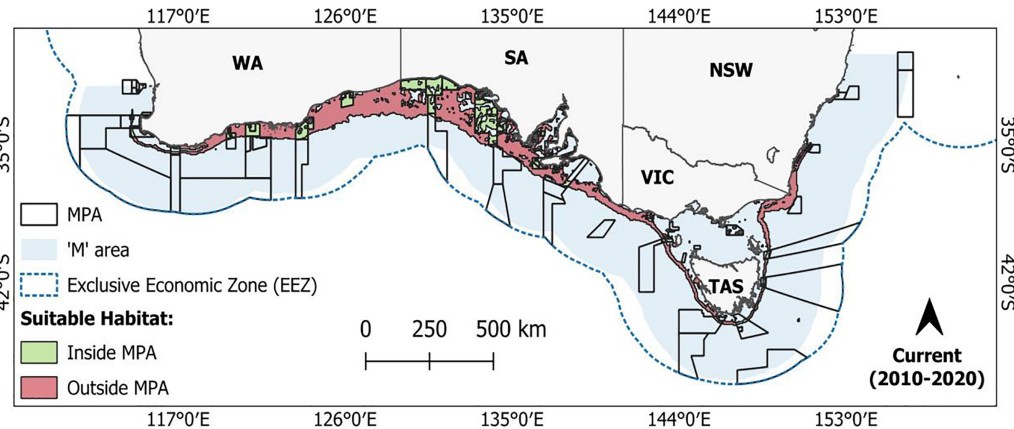

**Figure 5 Current (2010–2020) predicted suitable habitat for the whitefin swellshark (*Cephaloscyllium albipinnum*) inside (green) and outside (red) marine protected areas (MPAs).** Binary threshold = 0.3576.

(SS1-1.9 by 2100) projected a 4.7% (3,613 km²) increase in suitable habitat occurring within MPAs relative to the current distribution (Table 2; Fig. 6). All remaining scenarios, however, show a decline in suitable habitat occurring within an MPA. By the middle of the century (2050), these decreases range from 0.6% (476 km²) under the lowest emission scenario (SS1-1.9) to 34.5% (26,388 km²) under the highest emission scenario (SSP5-8.5)

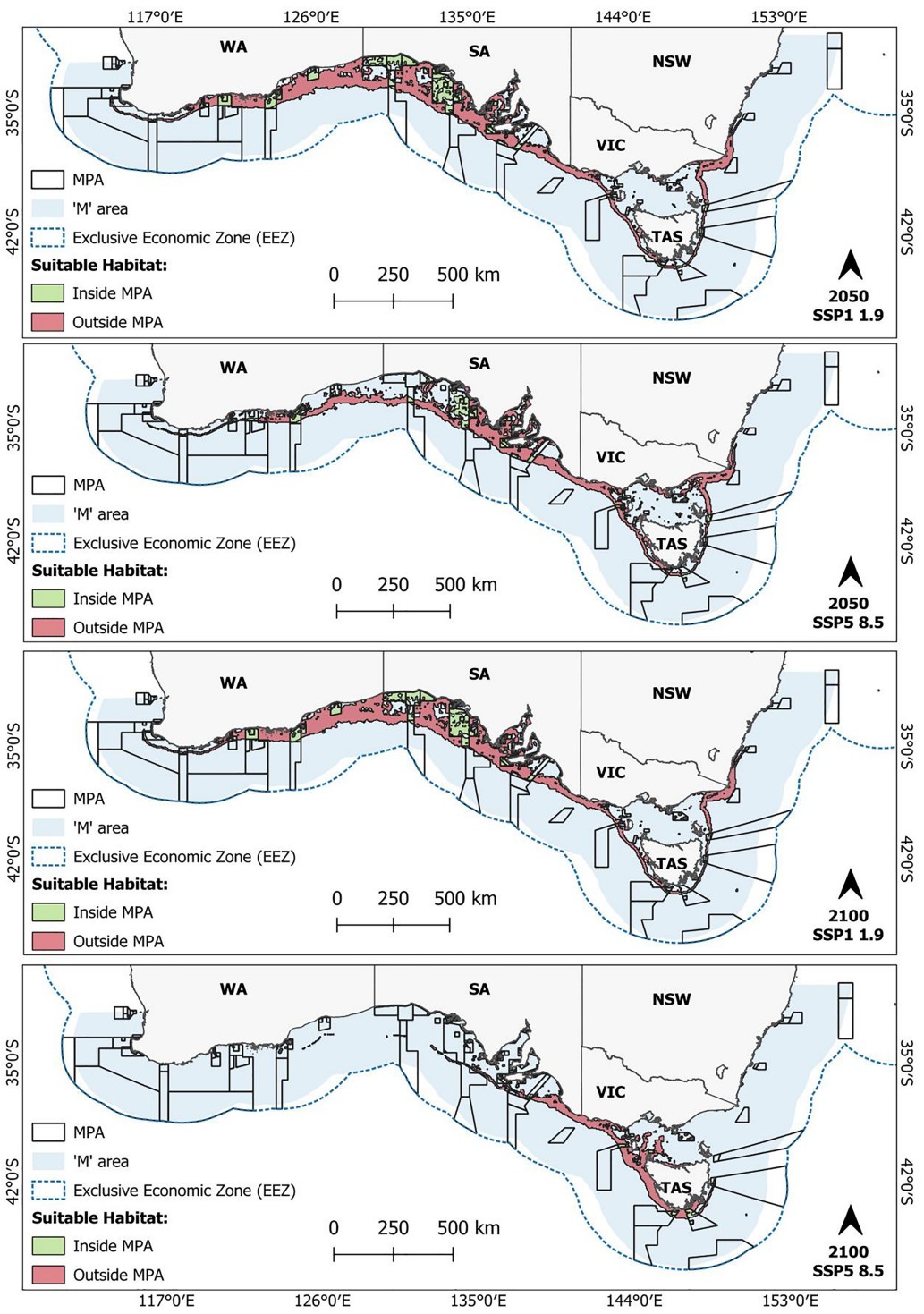

**Figure 6 Future predicted suitable habitat for the whitefin swellshark (*Cephaloscyllium albipinnum*) inside (green) and outside (red) marine protected areas (MPAs).** Suitable habitat predicted for scenarios SSP1-1.9 and SSP5-8.5 by the middle (2040–2050) (top two panels) and end of the century (2090–2100) (bottom two panels). Binary threshold = 0.3576.

(Table 2; Fig. 6; see Fig. S6). For the end of the century (2100), loss in coverage ranged from 25.3% (19,301 km$^2$) under SSP1-2.6 to a substantial 89.7% (68,475 km$^2$) under SSP5-8.5, leaving less than 2.5% (7,904 km$^2$) of suitable habitat remaining within MPAs (Table 2; Fig. 6; see Fig. S7). Most of this loss in MPA coverage is expected to occur within waters off WA and the majority of SA, as suitable habitat shifts from the South-west Marine Parks Network to the South-east Marine Parks Network, namely, the Huon Marine Park (1,748 km$^2$) and the Tasman Fracture Marine Park (1,412 km$^2$) (Fig. 6; Figs. S6 and S7). For a detailed breakdown of predicted suitable habitat overlap by IUCN category, see Table S3.

# DISCUSSION

## Habitat preferences

Benthic primary productivity was the most important factor influencing the distribution of *C. albipinnum*. Several biotic factors have previously been shown to influence movement in sharks, such as prey density and availability (*e.g.*, *Heithaus et al., 2002*) or predator avoidance (*Schlaff, Heupel & Simpfendorfer, 2014*). In marine ecosystems, the rate and distribution of primary production plays a fundamental role in structuring marine food webs (*Brown et al., 2010*). Thus, the use of primary productivity (and/or chlorophyll-a concentrations) as a proxy for prey availability has become a common approach in modeling the distribution of shark species (*Druon et al., 2022*; *Feitosa et al., 2020*; *Finucci et al., 2021*; *Reynolds et al., 2024*). Second to primary productivity was surface ocean temperature, another well-established driver in determining shark distribution (*Schlaff, Heupel & Simpfendorfer, 2014*). As an ectothermic species, *C. albipinnum* relies primarily on the surrounding environment to regulate its body temperature, directly influencing vital metabolic and physiological functions such as digestion, growth, and reproduction (*Bernal et al., 2012*). Numerous studies have reported the importance of prey availability and temperature in influencing shark distribution. For example, spatial patterns for blue sharks (*Prionace glauca*) (*Druon et al., 2022*), juvenile bull sharks (*Carcharhinus leucas*) (*Matich et al., 2024*), juvenile white sharks (*Carcharodon carcharias*) (*White et al., 2019*), and whale sharks (*Rhincodon typus*) (*Reynolds et al., 2024*) were all shown to be shaped by both ocean temperature and prey availability. Following primary productivity and surface ocean temperature, slope emerged as another significant factor influencing habitat suitability for *C. albipinnum*. Steeper slopes are often linked to underwater features such as seamounts, canyons, and continental slopes, which promote nutrient-rich upwelling currents. This increased nutrient availability attracts prey species, creating favorable feeding grounds for sharks (*Afonso, McGinty & Machete, 2014*; *Morato et al., 2010*).

While seawater velocity had less influence on the distribution of *C. albipinnum*, it plays a crucial role in shaping the spatial behaviour of other shark species. For example, grey reef sharks (*Carcharhinus amblyrhynchos*) have been found to utilise currents to reduce energy expenditure (*Papastamatiou et al., 2021*), while in the Gulf Stream, blue sharks (*P. glauca*) exploit the cores of anticyclonic eddies to forage on mesopelagic prey at otherwise inaccessible depths due to thermal constraints (*Braun et al., 2019*; *Druon et al., 2022*). Comparable behaviour has also been observed in white sharks (*C. carcharias*) (*Gaube et al., 2018*). Salinity had a negligible effect on the distribution of *C. albipinnum*, likely due to the

study area being characterised by a limited salinity gradient, with values ranging between ~34.8 and 38.5 PSS. However, as a stenohaline species, *C. albipinnum* is adapted to environments with relatively stable salinity levels, meaning that significant fluctuations in salinity could drive movement. Salinity, however, is likely to play a more significant role for nearshore species frequently exposed to freshwater runoff and its associated salinity variations (*e.g.*, *Heupel & Simpfendorfer, 2008*).

## Current predicted distribution of *C. albipinnum*

Under current climate conditions, the predicted suitable habitat for *C. albipinnum* closely aligns with its recognised known range (*CSIRO Marine and Atmospheric Research, 2012*; *International Union for Conservation of Nature, 2012*). The highest probability of presence was concentrated along the outer continental shelf and upper slope, consistent with the documented depth range (125 to 555 m) for *C. albipinnum* (*Ebert, Dando & Fowler, 2021*). Although, some overpredictions were identified, particularly in a region spanning approximately 850 km along the WA coastline, from Israelite Bay, WA (33.6°S, 123.9°E) to just above Cape Naturaliste, WA (33.5°S, 115.0°E). These discrepancies still align with the literature, with the occurrence of *C. albipinnum* in this region being recognised. For instance, *White & Moore (2024)* have suggested a westward range extension of approximately 950 km into southern WA waters towards Albany (33.5°S, 115.0°E) when compared to the IUCN Red List (*International Union for Conservation of Nature, 2012*) known range.

## Future predicted distribution of *C. albipinnum*

Many marine species are expected to undergo poleward range shifts due to climate change (*Melo-Merino, Reyes-Bonilla & Lira-Noriega, 2020*; *Robinson et al., 2017*). Supporting this notion, *Gervais, Champion & Pecl (2021)* revealed that, as anticipated, Australian marine species are indeed shifting their ranges, with 87.3% of 198 species from nine phyla exhibiting poleward redistributions. Our findings, however, revealed an east-southeast range shift for *C. albipinnum*; this projected east-southeast shift is likely a response to geographical and bathymetric constraints (*Kitchel et al., 2022*). As a demersal species inhabiting the outer continental shelf and upper slope at depths ranging from 125 to 555 m (*Ebert, Dando & Fowler, 2021*), being endemic to southeastern Australia, it is likely that the lack of continental shelf southward is limiting its dispersal directly poleward. Alternatively, mirroring that of terrestrial species that have shown shifts to higher elevations (*e.g.*, *Larsen, 2011*; *Neate-Clegg et al., 2021*), a direct poleward shift for *C. albipinnum* would require a vertical redistribution into deeper waters. While movement off the continental shelf into deeper waters is possible, with previous research documenting some marine species successfully redistributing into deeper water in response to climate change (*e.g.*, *Dulvy et al., 2008*), current knowledge on the depth tolerance limit of *C. albipinnum* remains unknown, making the feasibility of vertical adaptation unclear.

As emissions rise, habitat loss is projected to intensify over time, with *C. albipinnum* experiencing significant reductions in suitable habitat, especially by the end of the century, under the SSP5-8.5 scenario, where more than a third (72%) of its habitat could become

unsuitable. This observed decrease in suitable habitat for *C. albipinnum*, particularly under moderate and high emission scenarios, by the end of the century is similar to predictions for other shark species. For instance, *Birkmanis et al. (2020)* predicted an overall decrease in suitable habitat across the Australian exclusive economic zone (EEZ) for requiem sharks under two representative concentration pathway (RCP) emission scenarios (RCP4.5 and RCP8.5) by the end of the 21st century (2050–2099). Similarly, in the Gulf of Mexico under the RCP8.5 emissions scenario, *Braun et al. (2023)* observed a habitat loss of over 60% expected in the future (2070–2099) for the shortfin mako shark (*Isurus oxyrinchus*).

In contrast to more mobile species like the shortfin mako, *C. albipinnum* is less able to migrate to other suitable habitats, with the continental shelf surrounding Tasmania representing the southernmost shelf in Australia and sea surface temperatures projected to increase by 0.86 °C to 2.89 °C by the end of the century, depending on greenhouse gas emission levels (*Intergovernmental Panel on Climate Change, 2023*), remaining suitable habitat could approach the upper thermal tolerance of *C. albipinnum*. Should warming continue and render these regions uninhabitable, *C. albipinnum* would lack any further accessible shelf habitats to shift into, potentially leading to its extinction, unless *C. albipinnum* could physiologically adapt to remain in an altered environment. Acclimatisation, however, comes with an energetic cost, which can impact other functions such as growth, foraging, swimming, and reproduction (*Schlaff, Heupel & Simpfendorfer, 2014*). Tolerance and acclimation capacity, however, are species-specific, and previous research has shown both positive and negative impacts towards warming. For example, when exposed to projected end-of-century temperatures, *Gervais et al. (2018)* reported that juvenile epaulette sharks (*Hemiscyllium ocellatum*) exhibited significantly depressed growth rates and 100% mortality. Small-spotted catsharks (*S. canicula*) showed a temperature-induced increase in embryonic growth rate (*Musa et al., 2020*), while juvenile Port Jackson sharks (*Heterodontus portusjacksoni*) showed an increased learning performance; they also exhibited an increase in mortality (*Vila Pouca et al., 2019*). While our response curve indicated a relatively narrow thermal niche for *C. albipinnum*, with a suitable temperature range between 17 and 20 °C. critical knowledge of the thermal tolerance limit and acclimation capacity for *C. albipinnum* remains scarce.

## Conservation implications

To date, Australia has no specific conservation or management measures that exist for *C. albipinnum* (*Pardo et al., 2019*). However, it should be noted that at the time of writing, in 2019, identified by the National Environmental Science Program (NESP) Shark Action Plan (*Heupel et al., 2018*), *C. albipinnum* was nominated and is currently being considered for protection under the *Environment Protection and Biodiversity Conservation Act 1999* (*Federal Register of Legislation, 1999*). However, this process can take several years, with a further extension approved for 30th October 2025. Despite this, adequately implemented existing MPAs could contribute to the conservation of *C. albipinnum* through indirect benefits, such as habitat protection and reduced fishing pressure (*Albano et al., 2021*; *Speed, Cappo & Meekan, 2018*). Our results, however, revealed a concerning decline in spatial overlap between the predicted suitable habitat area for *C. albipinnum* and currently

designated MPAs. Coverage was lost across all future scenarios (except SSP1-1.9 by 2100), showing a general trend of reduced overlap as emissions increase and over time. Specifically, we found overlap between predicted suitable habitat for *C. albipinnum* and MPAs varied from 23.7% (76,379 km$^2$) for the current distribution to as little as 2.5% (7,904 km$^2$) coverage under the SSP5-8.5 scenario by 2100.

These findings, combined with the predicted shifts in suitable habitat moving in an east-southeast direction towards TAS and beyond jurisdictional borders, underscores the urgent need for adaptive management strategies to address the changing spatial dynamics of suitable habitat and ensure the conservation of *C. albipinnum*, including coordinated management efforts under the jurisdiction of both State and Commonwealth waters to effectively protect and conserve this Critically Endangered species throughout its range. As *Bowlby et al. (2024)* highlight, integrating habitat predictions into fisheries management frameworks can optimize spatiotemporal strategies, enhancing adaptive responses to climate-driven habitat shifts. Furthermore, our study demonstrated that refugia areas estimated suitable by all climatic projections will be of small size (45,990 km$^2$). This remaining patch will be a critical refuge for *C. albipinnum* and should be prioritised for targeted conservation efforts. Strengthening conservation measures in these areas is essential for the species' survival. Additionally, gains in suitable habitat are forecasted in regions like the Tasmanian continental shelf and Bass Strait, offering potential conservation planning and management opportunities.

## CONCLUSION

This study explores the impacts of climate change on the spatial distribution of the whitefin swellshark (*C. albipinnum*), finding that projected shifts in habitat suitability are possible in the future under various emission scenarios. While one scenario suggests an expansion of suitable habitat, particularly in response to lower greenhouse gas emissions, the overall trend points towards a contraction of habitat range, especially by the end of the century. Moreover, the observed east-southeast range shift towards TAS raises concerns for the species' long-term survival, particularly given the limited accessible shelf habitat in the region. Additionally, the decline in spatial overlap between suitable habitat and designated MPAs underscores the need for adaptive management strategies to conserve *C. albipinnum*. Urgent collaborative efforts, spanning both State and Commonwealth waters, are required to address the changing spatial dynamics of suitable habitat and mitigate the threats posed by climate change to this vulnerable species.

### Funding
The authors received no funding for this work.

### Competing Interests
The authors declare that they have no competing interests.

## Author Contributions

- Kerry Brown conceived and designed the experiments, performed the experiments, analyzed the data, prepared figures and/or tables, authored or reviewed drafts of the article, and approved the final draft.
- Robert Puschendorf conceived and designed the experiments, performed the experiments, analyzed the data, prepared figures and/or tables, authored or reviewed drafts the article, and approved the final draft.

## Data Availability

All data are available at https://www.bio-oracle.org/index.php (environmental/climate data) and GBIF for species localities.

The processing was done in R, the code is available in the Supplemental File.

The raw data used in the analysis is available in figshare: Puschendorf, Robert (2024). Raw data for 'Future climate-driven habitat loss and range shift of the Critically Endangered whitefin swellshark (*Cephaloscyllium albipinnum*)' by Kerry Brown and Rob Puschendorf. figshare. Dataset. https://doi.org/10.6084/m9.figshare.27215322.v1.

## Supplemental Information

Supplemental information for this article can be found online at http://dx.doi.org/10.7717/peerj.18787#supplemental-information.

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
