# Peer review of "Future climate-driven habitat loss and range shift of the Critically Endangered whitefin swellshark (Cephaloscyllium albipinnum)"

_PeerJ, doi:10.7717/peerj.18787_

## Round 0.1 · original submission · Major Revisions

Dear Authors,

We have received three reviews and all three reviewers agree on the importance of the paper and the results obtained. Furthermore, they gave great suggestions on how to increase the power of the work and how to improve it. All three reviewers agree to implement the methods and in particular, two of them suggest reanalyzing the data including biological factors and a new version of BIO-ORACLE. Finally, everyone agrees that the document needs a major revision. So please read all comments and suggestions carefully and provide us with an improved version of the manuscript.

Reviewer 1 ·

Basic reporting

This study titled "Projected Habitat Loss and Range Shift of the Critically Endangered Whitefin Swellshark (Cephaloscyllium albipinnum) in response to climate change" is well designed and the modeling is important to see whether swellshark will affect from climate change or not. The studies for such critical animals are valuable and it is obvious that they will fill an important gap in the literature.

Experimental design

I have a main question.
Why did not authors use Bio-ORACLE v3.0 (Assis et al., 2024)?This version was produced from the CMIP6 Earth system models, and for the future conditions, they have a lot of layers. The different layers can be shown different results for the model.

Assis, J., Fernández Bejarano, S.J., Salazar, V.W., Schepers, L., Gouvêa, L., Fragkopoulou, E., Leclercq, F., Vanhoorne, B., Tyberghein, L., Serrão, E.A., Verbruggen, H., De Clerck, O. (2024) Bio-ORACLE v3.0. Pushing marine data layers to the CMIP6 Earth system models of climate change research. Global Ecology and Biogeography. DOI: 10.1111/geb.13813.

Other comments:
Minor comments:
In Occurrence data section on line 113.
How many did you obtain occurrence data from GBIF? and after all filtration process, how many data have you continued to model? This status should be mentioned here.

On line 138
There is a new version for Bio-ORACLE.

On line 165
Do you have a citation for this threshold value?

On line 195
Why didn't you include other feature classes?

On line 221
Please rewrite "Were areas with......."

Validity of the findings

These findings are enough.

Reviewer 2 ·

Basic reporting

This paper aims to predict the distribution of an endemic shark species (the white shark, Cephaloscyllium albipinnum) in South Australia based on future climate predictions.
It is very well written and the topic may be of interest not only to specialists, but also to conservationists in general, since several marine species will face the same problems, such as habitat loss and climate dispersal.
The English is clear and unambiguous. Some comments have been added to make some technical statements clearer.
References to the literature are sufficient and the methodology is well explained, as are the codes used and data sourcing.
The table and the Figures are quite clear, but the tables in the supplementary materials need legends and some clarification.

Experimental design

The research is within the aims and scope of the journal, and a relevant question about the distribution of biodiversity, especially from the perspective of climate change, has been raised. This is especially true for endemics, such as the one reported in this paper.
Given the limited knowledge of the species, which is of relatively recent discovery, and covers a rather small range, the present work is helpful in filling this knowledge gap.
The method used is quite useful and it is explained quite well, however, I am quite surprised and disappointed that the authors did not consider biotic factors. In fact, it is known from literatures that biotic factors play a key role in the distribution of marine species, including sharks. Few references have been suggested.
With this in mind, I thought it wise to suggest a re-analysis, including biotic factors, or the inclusion of a little discussion about the limitations of using exclusively abiotic factors in such distribution predictions.
The information and the metadata and script attached with the manuscript are useful for the reproducibility of the present work.

Validity of the findings

The findings are quite useful, especially from the perspective of conservation and management of such endemic species.
In addition, the work raises the importance of a correct species management, including through the creation of Marine Protected Areas that take into account the climate changes we are facing.
Such changes underscore the need for a very rapid response, including bureaucratic-managerial, to better conserve marine biodiversity.
The results are statistically robust and well explained, few comments were made about the variables choice during the first stages of model creation.

Annotated reviews are not available for download in order to protect the identity of reviewers who chose to remain anonymous.

Reviewer 3 ·

Basic reporting

Grammar needs work throughout as does flow and order of information presented. Good level of background information provided and a solid set of figures and tables. Raw data is publicly available.
See details attached.

Experimental design

Research is original and within PeerJ scope, question is well defined in the introduction and knowledge gaps are suitably identified.
Methods are not described in sufficient detail.
See details attached.

Validity of the findings

Benefit to wider literature needs to be stated more clearly, conclusion could also be improved.
See details attached.

Additional comments

See details attached.

Annotated reviews are not available for download in order to protect the identity of reviewers who chose to remain anonymous.

---

## Round 0.2 · Minor Revisions

Dear Authors,

We have received comments from two reviewers who found the manuscript really improved compared to the previous version. In particular, they are happy with the improvement regarding the integration of biotic factors in the analyses and the open-science approach, which will make the codes available to replicate the analyses. They also give some minor comments and suggestions that will improve more the paper.

Reviewer 1 ·

Basic reporting

no comment

Experimental design

no comment

Validity of the findings

no comment

Additional comments

This has been improved well according to previous version. Therefore, I could not see any problem. It can be accepted to publish in this way.

Reviewer 2 ·

Basic reporting

The revised manuscript "Future climate-driven habitat loss and range shift of the Critically Endangered whitefin swellshark (Cephaloscyllium albipinnum)" is very well written and the authors made a huge work, integrating all the suggestions and edits of the reviewers, improving notably the already good work made.
The integration of the biotic factors in the analyses, as suggested in the previous round, have notably improved the manuscript, and brought out several points of discussion of considerable interest.
In the light of these new results, however, some minor points have been included in this second round of review. It is of particular importance to integrate the references section with some topic-specific, and more recent, references about the use of biotic factors in SDMs.
The general structure of the manuscript, the figures and the tables are clear and precise, helping the reader to go through the entire work without difficulties.
I also appreciate the open-science approach of the authors, making available the codes for reproducibility reasons.

All the specific comments are appended to the revised pdf in attachments, and here:

Line 103: Bottom longlines? It is better to clarify, since most of the time the term "longlines" is used for the pelagic fishing gears.

Lines 112-116: I think that habitat predictions may used also to inform fisheries management, since this would be less static than MPAs.

Lines 413-416: Glad to see that including biotic factors has an impact.

Lines 417-420: This is abosolutely and globally true.In the light of these new results, please include this more recent and global-scale study, where the authors used two environmental proxies for for food availability (chlorophyll-a gradient and upper mesopelagic micronekton) as indicators of preferred habitat in blue sharks:

Druon et al., (2022) Global-Scale Environmental Niche and Habitat of Blue Shark (Prionace glauca) by Size and Sex: A Pivotal Step to Improving Stock Management. Front. Mar. Sci. 9:828412. doi: 10.3389/fmars.2022.828412

Also, in the whale shark loss and gain of habitat suitability has been investigated for future scenarios, with an interesting focus on bioenergetics:

Reynolds et al., (2024). Science of The Total Environment, 951(2024): 175832. doi: 10.1016/j.scitotenv.2024.175832

Lines 420-422: Same as previous comment.

Lines 425-429: Here are some more recent and topic focused refs:

Druon et al., (2022) Global-Scale Environmental Niche and Habitat of Blue Shark (Prionace glauca) by Size and Sex: A Pivotal Step to Improving Stock Management. Front. Mar. Sci. 9:828412. doi: 10.3389/fmars.2022.828412

Reynolds et al., (2024). Science of The Total Environment, 951(2024): 175832. doi: 10.1016/j.scitotenv.2024.175832

White et al., (2019). Quantifying habitat selection and variability in habitat suitability for juvenile white sharks. PLoS ONE 14(5): e0214642. doi: 10.1371/journal.pone.0214642

Lines 440-442: This has been widely discussed in previously suggested recent reference, Druon et al., 2022.

Lines 447-449: Totally agree.

Lines 522-527: This is particularly true for such static species, and due to the "spatially fixed" nature of MPAs that cannot follow sharks in their habitat shifts. But it is true that SDMs and Habitat Predictions may serve as tools for better management of sharks. This is the case of better fisheries management informed by habitat predictions. Of course this may be included in a climatic scenario.
Such statement would enforce the conclusions of this interesting and important work. Consider to include this ref well expaining how SDM and Habitat Predictions may be important tools in this sense:

Bowlby et al., (2024). Global habitat predictions to inform spatiotemporal fisheries management: Initial steps within the framework. Marine Policy, 164:106155. doi: 10.1016/j.marpol.2024.106155.

Lines 533-538: As mentioned in the previous comment.

Experimental design

The work fits perfectly the aims and scope of the journal.
The research question is clear and the entire work is hypothesis-driven and it is well explained how this work will fill knowledge gap.
The totality of the analyses made are robust and reproducible, and the methods are well described

Validity of the findings

The finding of the manuscripts are of particular importance for the conservation of benthic sharks, which are underrepresented,in favour of more iconic species.
It is of extreme interest how this work may contribute to the study of less vagile species and how climate change may impact their survival, due to the less vagility, and then lower habitat shift potential, of these species.
The statistics are robust, and the avaibility of all the scripts made this work quite reproducible.
Conclusions are well stated.

Annotated reviews are not available for download in order to protect the identity of reviewers who chose to remain anonymous.

Reviewer 3 ·

Basic reporting

The article now meets all PeerJ standards.

Experimental design

The article now meets all PeerJ standards.

Validity of the findings

The article now meets all PeerJ standards.

Additional comments

I think the authors have done an excellent job of addressing extensive comments and, as such, the manuscript is greatly improved. I commend their efforts and appreciate the detailed response to the comments from round 1. The methods and results are clear and concise and now the readability, flow and impact has been maximised. I think it is very close to being ready for publication with only a few minor adjustments:

Line #69: Swap ‘owning’ to ‘owing’.
Line #91: Consider swapping ‘significant’ for ‘extensive’.
Line #1222: Consider this new reference for another example of a highly mobile shark species: https://www.nature.com/articles/s41558-024-02129-5#author-information
Line #132: This line doesn’t need a comma.
Line # 142: I would add ‘which are’ in front of ‘constraints’.
Line #158: Add ‘where’ before ‘localities’. Also should localities be occurrences?
Line #169: Do you mean 0.05 here?
Line #274: Can use ROC here as you have defined above.
Line #368: Add ‘falls’ (or similar) before ‘within’.
Line #496: Consider new sentence with ‘As surface temperatures...’

---

## Round 0.3 · accepted · Accept

Dear Authors, I have seen that you have integrated all the comments made by the referees in this new version, so I am happy to inform you that your paper can be accepted for publication.